# PGraphDTA: Improving Drug Target Interaction Prediction using Protein Language Models and Contact Maps

**Rakesh Bal**  **Yijia Xiao**  **Wei Wang**

Dept. of Computer Science, University of California Los Angeles

{rakesh.bal, yijia.xiao, weiwang}@cs.ucla.edu

## Abstract

Developing and discovering new drugs is a complex and resource-intensive endeavor that often involves substantial costs, time investment, and safety concerns. A key aspect of drug discovery involves identifying novel drug-target (DT) interactions. Existing computational methods for predicting DT interactions have primarily focused on binary classification tasks, aiming to determine whether a DT pair interacts or not. However, protein-ligand interactions exhibit a continuum of binding strengths, known as binding affinity, presenting a persistent challenge for accurate prediction. In this study, we investigate various techniques employed in Drug Target Interaction (DTI) prediction and propose novel enhancements to enhance their performance. Our approaches include the integration of Protein Language Models (PLMs) and the incorporation of Contact Map information as an inductive bias within current models. Through extensive experimentation, we demonstrate that our proposed approaches outperform the baseline models considered in this study, presenting a compelling case for further development in this direction. We anticipate that the insights gained from this work will significantly narrow the search space for potential drugs targeting specific proteins, thereby accelerating drug discovery. Code and data for PGraphDTA are available at `https://github.com/Yijia-Xiao/PGraphDTA/`.

## 1 Introduction

Understanding the binding affinity between a drug and its target protein can help predict the potential efficacy of a drug candidate and guide the development of new drugs. Existing approved drugs can also be verified for new diseases to check if they offer a promising solution. This is called drug repurposing and it accelerates the drug discovery process significantly. Hence, it is of paramount importance to validate the behavior of several drugs against a target protein to find a suitable candidate best suited to solve diseases involving that protein DiMasi et al. [2003]. However, finding the binding affinity using laboratory experimental methods is expensive and is a deterrent to testing many drugs as potential candidates for binding with important targets. Several computational methods Corsello et al. [2017], Iskar et al. [2012] have been proposed to help solve this issue faster and reduce the search space for reaching the right drug(s) for the target(s).

Recently, a variety of deep learning based methods have been proposed (Nguyen et al. [2021], Öztürk et al. [2018], Jiang et al. [2020]) to solve this problem. These methods provide a quick framework to predict the binding affinities between drugs and their targets, and they are shown to provide state-of-the-art performance in many open datasets. However, many of these datasets are small, structurally homogeneous Davis et al. [2011] and hence prohibit the ability of the models trained on them to generalize to unseen drug-target pairs. We aim to study these problems and propose easy-to-use improvements to a multitude of model architectures.

NeurIPS 2023 AI for Science Workshop.

| Paper | Model | Drug Representation | Target Representation | Drug Model | Target Model |
|---|---|---|---|---|---|
| Öztürk et al. [2018] | DeepDTA | SMILES | Sequence | CNN | CNN |
| Öztürk et al. [2019] | WideDTA | SMILES & LMCS $^{\beta}$ | Sequence & PDM $^{\dagger}$ | CNN | CNN |
| Nguyen et al. [2021] | GraphDTA | SMILES $\rightarrow$ Graph | Sequence | **GNN** | CNN |
| Jiang et al. [2020] | DGraphDTA | SMILES $\rightarrow$ Graph | Sequence $\rightarrow$ Graph | **GNN** | **GNN** |

Table 1: Survey Table on Related Models. $^{\dagger}$ *PDM-* Protein Domains and Motifs, $^{\beta}$*LMCS* - Ligand Maximum Common Substructures

Building upon the foundation laid by existing research, particularly GraphDTA Nguyen et al. [2021], our work seeks to enhance performance by implementing strategies that address inherent limitations. These include the inability to adequately capture essential interactions between protein and drug embeddings and the insufficient representation of inductive bias in sparse datasets.

The contributions of this work can be summarized into:

1. A thorough survey of prior research in Drug-Target Interaction (DTI), highlighting various methodologies employed. We delve deep into the recent achievements in Protein Language Models, leveraging these advancements to enhance the performance of DTI predictions.

2. A shift from utilizing Convolutional Neural Networks (CNNs) in recent DTI models for protein sequence modeling to deploying pretrained Protein Language Models (PLMs). Our results indicate that PLMs offer a more effective representation of Amino Acid sequences, yielding enhanced binding affinity predictions.

3. An exploration of techniques to discern interatomic distances between atoms and amino acids. By integrating this information into our model architecture, we demonstrate superior prediction outcomes. Particularly, the inclusion of inductive bias proves beneficial for performance on smaller datasets like DAVIS, equipping the model with essential data to refine binding affinity predictions.

## 2   Related Works

### 2.1   Drug Target Interaction (DTI) Models

The development of a new drug can demand investments reaching several billion US dollars and may necessitate over a decade to secure FDA approval Ashburn and Thor [2004], Roses [2008]. Given the protracted and expensive nature of drug development, there is an intensified drive to devise computational models capable of predicting the binding affinity of novel drug-target pairs. A multitude of computational methodologies have been introduced for DTI prediction Corsello et al. [2017], Cao et al. [2012], Cao et al. [2014]. One popular technique followed by researchers is Molecular Docking which involves predicting the stable 3D configuration of the drug-target complex through a scoring function Li et al. [2019]. *SimBoost* employs collaborative filtering to discern similarities in the affinity of various drugs and targets, thereby constructing features for new pairs He et al. [2017]. Several Kernel-based strategies leverage molecular descriptors of drugs and targets within a Regularized Least Squares Regression (RLS) framework Cichonska et al. [2017], Cichonska et al. [2018].

DeepDTA Öztürk et al. [2018] was one of the first deep-learning based DTI models to successfully model the DTI prediction problem. DeepDTA used Convolutional Neural Networks (CNNs) to encode proteins and drugs, setting a new standard in predicting binding affinity. It outperformed preceding non-deep learning models, laying a foundation for subsequent advancements in the field. Building upon DeepDTA's groundwork, WideDTA Öztürk et al. [2019] encapsulated drug and protein sequences into higher-order features. In this model, the drugs are represented by the most common sub-structures (the Ligand Maximum Common Substructures (LMCS) Woźniak et al. [2018] and the proteins are represented by the most conserved sub-sequences (the Protein Domain profiles or Motifs (PDM) from PROSITE Sigrist et al. [2010].

(a) Clc1ccc(Nc2nnc(Cc3ccncc3) c3ccccc23)cc1

(b) O=C(NC1CCNCC1)c1[nH] ncc1NC(=O)c1c(Cl)cccc1Cl

Figure 1: Example of SMILES string and their corresponding molecule

Drawing inspiration from both DeepDTA and WideDTA, GraphDTA Nguyen et al. [2021] represented drugs as graphs and harnessed Graph Neural Networks (GNNs) to enhance drug representations, achieving superior performance and outshining DeepDTA in key datasets. Later, DGraphDTA Jiang et al. [2020] took this a step further by converting proteins into graph structures via contact maps, employing GNNs for both drugs and targets. This comprehensive GNN-based approach further refined prediction capabilities, setting a new benchmark surpassing GraphDTA's performance. A tabular form of the various representations and models used in these models is presented in Table 1.

## 2.2 Graph Neural Networks (GNNs)

Graph Neural Networks (GNNs) have gained substantial attention in recent years due to their ability to effectively model and analyze complex structured data, such as social networks, biological networks, and recommendation systems. In this field, Graph Convolutional Networks (GCNs) Kipf and Welling [2016] was one of the first architectures to gain widespread acclaim as they successfully trained a GNN using the concept of convolution inspired by CNNs. Their simplicity proved them very useful in a variety of tasks like link prediction and node classification. Similarly, Graph Attention Networks Veličković et al. [2017] were able to model Graph Networks with attention which provided a more accurate representation of graphs. Numerous variants of Graph Networks are used in a wide variety of tasks today.

## 2.3 Protein Language Models (PLMs)

In recent years, attention-based deep learning models built on the Transformer architecture Vaswani et al. [2017] have become state-of-the-art across natural language processing (NLP), computer vision, and other AI domains. Language models like BERT Devlin et al. [2018], T5 Raffel et al. [2020], and GPT-3 Brown et al. [2020], which use Transformer encoders, have achieved strong performance on a wide range of NLP benchmarks Kalyan et al. [2021]. Similarly, Vision Transformer (ViT) models Dosovitskiy et al. [2020], which apply Transformer encoders to image data, have surpassed convolutional neural networks (CNNs) on computer vision tasks Khan et al. [2021]. The self-attention mechanism underpinning Transformers allows modeling complex global dependencies in data, leading to their rapid adoption over previous state-of-the-art models across modalities.

Protein Sequences can be visualized as sentences, with each amino acid residue being equivalent to a word. This analogy paves the way for the application of language models, which are ubiquitous in Natural Language Processing (NLP), to model proteins. A variety of Protein Language Models (PLMs) have recently been introduced, including, but not limited to, ProtBERT Brandes et al. [2021], ProtT5 Elnaggar et al. [2020], ProteinLM Xiao et al. [2021], and ProGen Madani et al. [2023]. These protein language models share the same Transformer architecture. DistilProtBERT Geffen et al. [2022] is a distilled version of ProtBERT inspired from DistilBert Sanh et al. [2019]. ESM Rives et al. [2021], and ESM2 Lin et al. [2023] are another branch of PLMs that provides precise representations of proteins based on evolutionary information. SeqVec Heinzinger et al. [2019] is a PLM that, instead of being based on Transformers, is inspired by ELMo Peters et al. [1802] and provides fast and efficient representations of protein sequences. All of these models are pre-trained on large protein datasets like UniRef-50 Suzek et al. [2015] and then finetuned on downstream tasks. We leverage

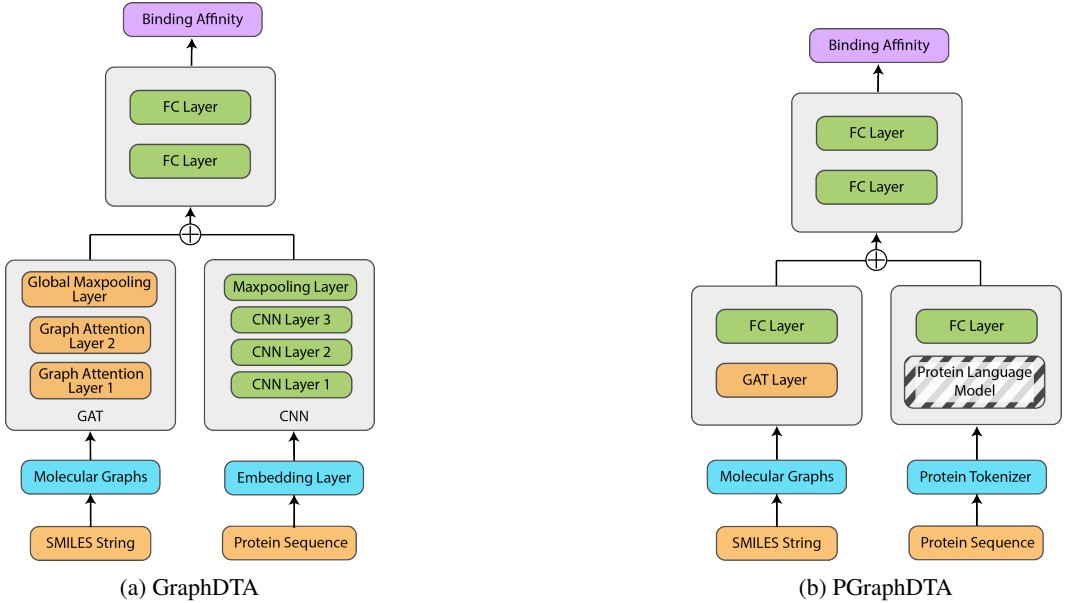

(a) GraphDTA            (b) PGraphDTA

Figure 2: Baseline GraphDTA and PGraphDTA model architecture. In *(b)*, dashed lines indicate PLM embeddings are precomputed and are not a part of the training loop.

these models in this work because of their exceptional performance in many downstream tasks. PLMs with their parameters and datasets on which they are trained on is presented in Table 2.

## 3 Materials and Methods

### 3.1 Drug Representation

We use the Simplified Molecular Input Line Entry System (SMILES) to represent molecules in a simple format to be readable by computers Weininger [1988]. This format allows for fast retrieval and substructure searching. The SMILES code is a string that can be processed using several Natural Language Processing (NLP) techniques or by CNN layers. In this work, we treat the drug compounds as graphs where the nodes represent the atoms and edges represent their interactions. We transform the SMILES string into a molecular graph and derive atomic features using the **RDKit** Landrum [2016]. Representative SMILES strings are illustrated in Figure 1.

### 3.2 Protein Representation

Proteins are represented as Amino Acid (AA) sequences which makes them analogous to sentences in NLP, with each AA equivalent to a word. As highlighted in the preceding section, strategies effective in NLP can also be harnessed to model these protein sequences. Notably, models such as DeepDTA Öztürk et al. [2018] and GraphDTA Nguyen et al. [2021] use CNN layers to this end.

In this work, we employ PLMs introduced in section 2.3 to encode the protein sequences. Each sequence is preprocessed to a maximum length by truncating them in case of longer sequences and padding them in case of shorter sequences. To ensure efficient processing, we preprocess sequences undergo preprocessing to achieve a uniform length longer sequences are truncated while shorter ones are padded. This step is crucial as it greatly reduces the computational overhead required to extract embeddings, as the PLMs can encode protein sequences in batches.

In our selection of PLMs, we focused on models that are both publicly available and have demonstrated efficacy in protein representation. Recognizing the versatile application of BERT models in various NLP tasks, we opted for ProtBERT Elnaggar et al. [2020] as one of the PLMs in our study. Additionally, DistilProtBERT Geffen et al. [2022] was chosen to explore the potential of distilled versions of large PLMs in the context of Drug-Target Interaction (DTI) prediction. ESM-2 Lin et al. [2023] was included due to its proven capability in providing superior representations across a diverse array of proteins and tasks. To assess the impact of non-Transformer based PLMs,

| Paper | PLM | Datasets Trained on | Number of Parameters | CASP12 | CASP14 |
|---|---|---|---|---|---|
| Heinzinger et al. [2019] | SeqVec | UniRef50 | 93M | 0.73 | — |
| Elnaggar et al. [2020] | ProtBERT | BFD100 UniRef100 | 420M | 0.75 | — |
| Geffen et al. [2022] | DistilProtBERT | UniRef50 | 230M | 0.72 | — |
| Lin et al. [2023] | ESM2 | UniRef50 | 650M | — | 0.51 |

Table 2: Survey Table on Protein Language Models used in this work

| Number | Dataset | Proteins | Compounds | Binding Entries |
|---|---|---|---|---|
| 1 | DAVIS | 442 | 72 | 25,772 |
| 2 | KIBA | 229 | 2068 | 117,657 |

Table 3: Dataset Statistics

we also incorporated SeqVec Heinzinger et al. [2019], which is grounded in the ELMo architecture, employing bidirectional LSTMs. This varied selection of models allows for a comprehensive analysis of different approaches to protein sequence representation in DTI prediction.

### 3.3 Datasets

The benchmark datasets proposed by DeepDTA are used for performance evaluation:

1. DAVIS Davis et al. [2011]: Contains the interaction of 72 kinase inhibitors with 442 kinases covering $> 80\%$ of the human catalytic protein kinome. We transform the binding affinity to log scale for stable training.

2. KIBA Tang et al. [2014]: Originated from an approach in which kinase inhibitor bioactivities from different sources such as $K_i$, $K_d$, and $IC_{50}$ were combined. This is a much larger dataset than DAVIS and contains more varieties of proteins than just the kinases.

Both DAVIS and KIBA are publicly accessible and can be downloaded using the pyTDC loader Huang et al. [2021]. Each drug-target interaction in these datasets is presented as a pair comprising a drug, denoted by a SMILES string, and a protein represented as a sequence. Detailed statistics for these datasets can be found in Table 3.

### 3.4 Models

In this work, we use GraphDTA Nguyen et al. [2021] as inspiration for our models and also as a baseline. Figure 2a illustrates the architecture of GraphDTA, which comprises two components: a GNN for encoding the drug represented as a graph and a CNN for encoding the protein sequence (amino acid sequence). These two components are subsequently concatenated and fed into two fully connected layers to predict the binding affinity value. Our experimentation builds upon this framework, utilizing its structural blueprint while introducing our proposed enhancements. Specifically, we use the GAT network for modeling the drugs and implement two distinct architectural modifications to enhance predictive accuracy. Both of these approaches are discussed below.

#### 3.4.1 PGraphDTA (Replacing CNNs with PLMs)

As discussed in section 2.3, PLMs have been shown to outperform other methods in modeling protein sequences. Therefore, we replaced the CNN encoder used in GraphDTA for protein sequence modeling with several PLMs, including ProtBERT, DistilProtBERT, ESM-2[1], and SeqVec. To improve computational efficiency, we precomputed the PLM embeddings for each protein sequence during preprocessing and cached these embeddings during training. This approach is equivalent to freezing the PLMs during training, substantially reducing GPU memory requirements and enabling

---

[1]For ESM2, we used the model with 650M parameters due to its compact size and efficiency

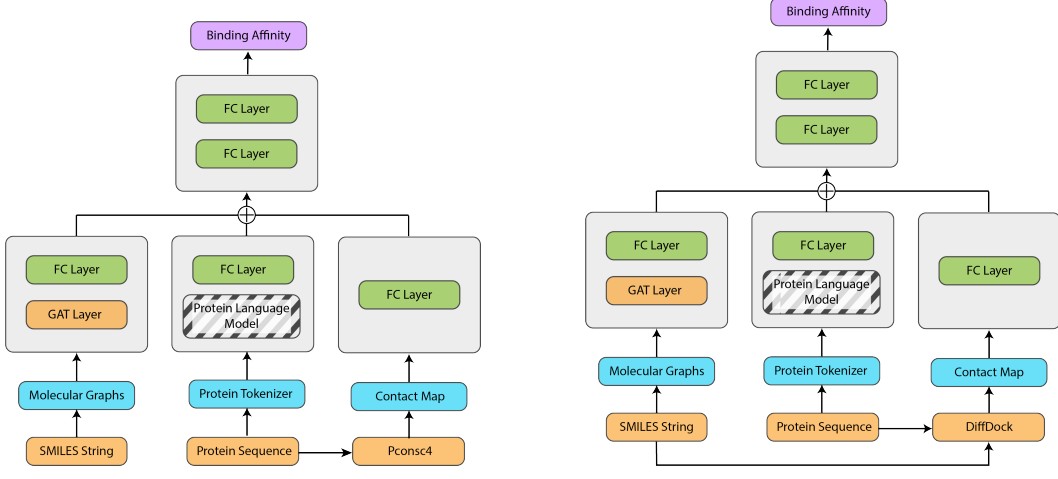

(a) Protein Contact Map using Pconsc4      (b) Molecular Contact Map using DiffDock

Figure 3: Model with Contact Map information. PLM embeddings are precomputed and, hence, is not a part of the training loop, as indicated by the dashed lines.

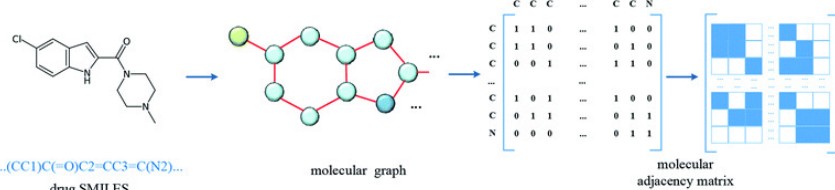

Figure 4: Molecular contact map prediction workflow, Source: DGraphDTA Jiang et al. [2020]

faster training with improved performance. Using this strategy, we trained our models for 1500 epochs, achieving enhanced results compared to the CNN encoder baseline. The overall architecture incorporating PLMs is depicted in Figure 2b.

### 3.4.2 PGraphDTA-CM (Adding Contact Maps information)

Using two approaches, we integrate contact maps, which are inadvertently intermolecular distance information, into the PGraphDTA model as inductive bias. Since the datasets used are relatively small, the inductive bias injected into the model, in theory, should help improve the model's performance.

**PGraphDTA-CM1**: In the first approach, we obtain intermolecular distances between binding sites of protein with drugs and create a contact map which would act as an additional source of information to our model to better predict the binding affinity. For this, we follow molecular docking, where we use DiffDock Corso et al. [2022] to predict the docking of the small drug with the protein. DiffDock is a molecular docking model that predicts the docking sites between proteins and drugs using diffusion and achieves state-of-the-art results. If provided, it processes the protein structure in PDB format. Otherwise, it uses ESMFold to predict the protein structure using the protein sequence. Next, we obtain the intermolecular distances between the different atoms in the small drug. We then construct an adjacency matrix of the intermolecular distances, apply a threshold of $10A^0$ to transform it into a binary matrix, and integrate this as an additional data source in our model, along with protein and drug encodings. An illustration of this workflow using DiffDock is shown in Figure 4.

**PGraphDTA-CM2**: In the second approach, we obtain the protein contact map, an adjacency matrix constructed from individual amino acids representing whether they are in contact with each other. Similar to 3.4.2, we expect the protein contact map to assist the PGraphDTA model to better assimilate the binding information and improve the overall performance. For a protein sequence of length $L$, the resulting contact map $M$ is a matrix with $L$ rows and $L$ columns, where each element $m_{ij}$ of $M$ indicates whether the corresponding residue pair (residue $i$ and residue $j$) are in contact. Two residues are considered to be in contact if the Euclidean distance between their $C_\beta$ is less than a

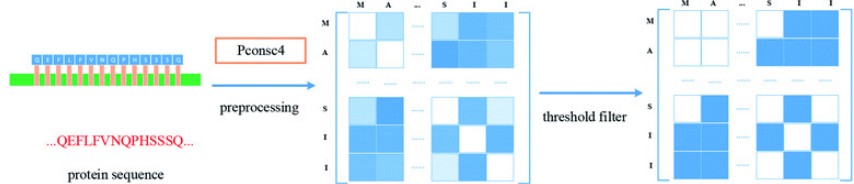

Figure 5: Protein contact map prediction workflow, Source: DGraphDTA Jiang et al. [2020]

| Model | MSE | |
| --- | --- | --- |
| | *DAVIS* | *KIBA* |
| GraphDTA (Baseline) | 0.271 | 0.205 |
| *PGraphDTA* (Ours) | | |
| – DistilProtBERT | 0.259 | 0.198 |
| – ProtBERT | 0.269 | 0.197 |
| – SeqVec | **0.221** | **0.193** |

Table 4: MSE Scores of different PLMs in PGraphDTA on both datasets

| Model | DAVIS | |
| --- | --- | --- |
| | **PGraphDTA** | |
| | *CM1* | *CM2* |
| DistilProtBERT | 0.270 | 0.228 |
| ProtBERT | 0.279 | 0.234 |
| ESM2 | **0.230** | **0.224** |

Table 5: MSE Scores of adding Contact Map information on DAVIS

specified threshold. Inspired by DGraphDTA Jiang et al. [2020], we use Pconsc4 Bassot et al. [2019] to predict the contact map with a threshold of $0.5$. Pconsc4 uses a U-net architecture Ronneberger et al. [2015], which operates on the 72 features calculated from each position in the multiple sequence alignment. This is similar to PGraphDTA-CM1 but is implemented for proteins using pconsc4, as shown in Figure 5. A comprehensive depiction of the model architectures for both methodologies is provided in Figure 3.

### 3.5 Evaluation Metric

We use Mean Squared Error (MSE) to evaluate our model's performance. MSE is a widely recognized metric for assessing the accuracy of regression models and is defined as:

$$MSE(y_i, \widetilde{y}) = \frac{1}{N} \sum (y_i - \widetilde{y})^2 \tag{1}$$

where $N$ is the number of samples in the dataset, $y_i$ represents the actual values of the target variable and $\widetilde{y}$ is the predicted value. MSE provides a measure of the overall accuracy of the regression model by quantifying the average magnitude of the squared errors.

## 4 Results & Discussion

### 4.1 PGraphDTA

Table 4 presents the results obtained using our proposed PGraphDTA model with various PLMs on the DAVIS and KIBA benchmark datasets. To establish a baseline, we reimplemented GraphDTA using its official repository[2]. All incorporated PLMs yield an improved MSE compared to the CNN encoder, with maximum enhancements of **18.45%** on DAVIS and **5.85%** on KIBA. The greater relative gains on DAVIS likely arise from its smaller size, as PLMs can better exploit the limited training data. These results demonstrate that PLMs more effectively represent protein sequences for binding affinity prediction than CNNs, especially when data is scarce. Our proposed approach could thus enable more accurate and robust predictions in settings with small datasets.

### 4.2 PGraphDTA-CM

**PGraphDTA-CM1:** The results obtained after incorporating contact maps information into our model are presented in Table 5. Interestingly, the PGraphDTA-CM1 variants of DistilProtBERT and ProtBERT performed worse than the baseline GraphDTA. Even the top-performing ESM2 based

---

[2]https://github.com/thinng/GraphDTA

PGraphDTA-CM1 underperformed its PGraphDTA counterpart. A likely explanation is that the interatomic distance maps generated by DiffDock contain substantial noise and inaccuracies for many examples, providing low-quality training data. This hypothesis is supported by DiffDock's reported top-1 success rate of only 38% ($RMSD < 2A^0$) on the PDBBind benchmark dataset Corso et al. [2022]. Despite representing the current state-of-the-art in molecular docking, DiffDock struggles to produce robust predictions on many docking tasks. Our findings highlight the persistent challenges and bottlenecks in molecular docking, even for sophisticated methods like DiffDock. Future work should explore approaches to denoise or improve the quality of predicted distance/contact maps to better exploit this structural information.

**PGraphDTA-CM2:** In contrast to PGraphDTA-CM1, the PGraphDTA-CM2 model incorporating protein contact maps from Pconsc4 yields improved performance over PGraphDTA for ProtBERT and DistilProtBERT. These results demonstrate that the Pconsc4-predicted contact maps provide higher-quality structural information than the DiffDock-generated distance maps for binding affinity prediction. The contact maps generated with Pconsc4 are much faster and less resource intensive than performing full molecular docking with DiffDock. Our findings highlight the promise of predicted contact maps as an efficient source of structural insight that can enhance performance when incorporated into DTI architectures. Further research on optimal integration strategies to leverage predicted contacts could lead to additional gains.

Across both PGraphDTA-CM model variations, **ESM2** consistently achieved the lowest MSE, demonstrating its superiority in representing protein sequences for this task compared to the other examined PLMs. Due to resource constraints, we were unable to report PGraphDTA-CM results on the larger KIBA dataset, which remains an avenue for future work. In summary, our results clearly highlight two effective strategies for improving binding affinity prediction in GNN/CNN based architectures for drug-target interaction. First, replacing CNN encoders with PLMs provides an efficient way to boost performance, even without fine-tuning. Second, incorporating predicted protein contact maps as an inductive structural bias can greatly enhance results, especially on small datasets where data is scarce.

## 5 Conclusion & Future Works

In this work, we critically examined recent DTI prediction models, analyzing their architectures and representations of proteins and drugs. We devised multiple approaches to improve the performance of the existing model architectures. Based on our results, we propose two model designs (*PGraphDTA* and *PGraphDTA-CM2*) with improvements that involve replacing the CNNs with the PLMs and adding the contact map information to the existing models, respectively. Both techniques yielded substantial gains, even with simple integration into the GraphDTA model architecture. Our findings highlight the potential of these computationally inexpensive modifications to boost binding affinity predictions across diverse model architectures. Several promising directions emerge for future work:

1. Evaluating our enhanced models on larger benchmarks like BindingDB [Liu et al. [2007]] would reveal their robustness and extensibility to new drug-target pairs.
2. Inspired from the PigNet architecture Moon et al. [2022], incorporating physics-based inductive biases like van der Waals forces, hydrogen bonding, and hydrophobic interactions could provide useful constraints and regularization when data is scarce.
3. Exploring cross-attention mechanisms between drug and protein embeddings [Vaswani et al. [2017], Hou et al. [2019]] may strengthen representation learning by capturing interactions between binding sites. Efficient implementations will be key for feasibility.

In conclusion, this work introduces impactful techniques for DTI prediction using protein language models and contact maps. Our integration strategies and analysis of diverse inductive biases establish a strong foundation for advancing this field. Continued exploration of representations, multimodal architectures, and physics-informed models offers tremendous potential to overcome challenges in drug discovery. By enhancing generalization to novel drug-target combinations, DTI prediction can help unlock new therapeutics.

## Acknowledgments and Disclosure of Funding

We acknowledge the initial support from AWS MLSL team comprising Erika Pelaez Coyotl, Ryan Brand, and Liwen You in the methodologies and compute resources.

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
