# OpenReview forum: "PGraphDTA: Improving Drug Target Interaction Prediction using Protein Language Models and Contact Maps"
_NeurIPS.cc/2023/Workshop/AI4Science — NeurIPS2023-AI4Science Poster_

### Official Review · Reviewer_9yZN · 2023-10-25
**Clarity and justification are needed for employing protein language models and contact maps**

**Rating:** 6
**Confidence:** 3

**Review:**

The authors proposed a method for predicting drug-target interactions utilizing protein language models and contact maps, building upon the GraphDTA framework. Several concerns have been raised about the paper, and the following points should be addressed to improve clarity and completeness:

1. Rationale for Choosing Protein Language Models:
The paper lacks an explanation regarding the specific reasons for choosing ProtBERT, DistilProtBERT, ESM-2, and SeqVec as pre-trained protein language models (as mentioned in lines 137-138). Providing a brief rationale for selecting these models would enhance the understanding of the authors' choice. Additionally, more protein language models should be employed.
2. Specification of ProtBERT Model:
Regarding ProtBERT, it is essential to specify which specific model variant of ProtBERT was employed in this study. Different versions and sizes of pre-trained models might have varying performance, and specifying the exact version used is crucial for reproducibility and comparison with future studies.
3. Reason for Exclusion of ESM2-650M and SeqVec in Tables:
The exclusion of ESM2-650M from Table 3 and SeqVec from Table 4 requires clarification. Providing reasons for not including these models in the tables would strengthen the methodology's transparency and aid readers in understanding the authors' choices.
4. Integration of Contact Map Information:
The paper mentions a puzzling finding: in Table 4, results with added contact map information were lower than without it. It is crucial to elucidate the main reasons for this unexpected result and to elaborate on the motivation behind integrating contact map information. Understanding the rationale behind this integration will help readers grasp the authors' experimental design and intentions more clearly.